# Formulation and Characterization of Sustainable Algal-Derived Nanoemulgels: A Green Approach to Minimize the Dependency on Synthetic Surfactants

**DOI:** 10.3390/polym16020194

**Published:** 2024-01-09

**Authors:** Patricia Tello, Jenifer Santos, Nuria Calero, Luis A. Trujillo-Cayado

**Affiliations:** 1Departamento de Ingeniería Química, Escuela Politécnica Superior, Universidad de Sevilla, c/Virgen de África, 7, 41011 Sevilla, Spain; pattelriv@alum.us.es; 2Departamento de Ciencias de la Salud y Biomédicas, Facultad de Ciencias de la Salud, Universidad Loyola Andalucía, Avda. de las Universidades s/n, 41704 Dos Hermanas, Sevilla, Spain; 3Departamento de Ingeniería Química, Facultad de Química, Universidad de Sevilla, c/Profesor García González s/n, 41012 Sevilla, Spain; nuriacalero@us.es

**Keywords:** biopolymer, phycocyanin, nanoemulgels, microfluidization

## Abstract

Phycocyanin (PC), a natural protein that is very interesting from the medical point of view due to its potent antioxidant and anti-inflammatory properties, is obtained from algae. This compound is gaining positions for applications in the food industry. The main objective of this work was to obtain nanoemulgels formulated with PC and k-carrageenan (a polymer that is obtained from algae as well). An optimization of the processing parameters (homogenization pressure and number of cycles) and the ratio of PC and a well-known synthetic surfactant (Tween 80) was developed using response surface methodology. The results of this optimization were 25,000 psi, seven cycles, and a 1:1 ratio of PC/Tween80. However, the necessity for the incorporation of a polymer that plays a thickener role was observed. Hence, k-carrageenan (k-C) was used to retard the creaming process that these nanoemulsions suffered. The incorporation of this biopolymer provoked the creation of a network that showed gel-type behavior and flow indexes very close to zero. Thanks to the combined use of these two sustainable and algae-obtained compounds, stable nanoemulgels were obtained. This work has proved that the combined use of PC and k-C has emerged as a sustainable alternative to stabilize dispersed systems for the food industry.

## 1. Introduction

Emulsions, a well-known dispersed system, have numerous applications in food, cosmetics, agrochemicals, and pharmaceutics. One of their most important characteristics is their physical stability. It is crucial to optimize not only the formulation but also the processing. Interestingly, there is a high-energy method that has shown the best results concerning the reduction in the droplet size: microfluidization. This method, which is based on passing the emulsion through nanometric channels at very high shear, is attracting a lot of attention to be applied in emulsions [1,2,3,4]. Surfactants, which play a crucial role in stabilizing emulsions, are commonly employed to lower the interfacial tension between immiscible phases and protect the interface from destabilization processes. However, the widespread use of synthetic surfactants has raised concerns due to their potential environmental and health impacts, prompting researchers and industries to explore alternative, more sustainable approaches. Reducing synthetic surfactants in emulsions has become an increasingly critical aspect of modern materials science and industrial processes [5,6].

Phycocyanin (PC), a natural protein mainly obtained from blue-green algae, has garnered significant attention across diverse fields, including medicine, nutrition, and cosmetics, due to its potential health benefits. Its potent antioxidant and anti-inflammatory properties make it a promising candidate for therapeutic applications. In the food industry, phycocyanin is employed as a natural blue food coloring, offering a healthier alternative to synthetic dyes since it is considered non-toxic and biocompatible. PC has been used as a stabilizer in emulsions [7,8] since it can reduce interfacial tension and prevent coalescence with the aging time [9,10]. However, its use has always been in combination with other stabilizers such as pectin, chitosan, or gelatine [8,10,11]. Its use with a synthetic surfactant, like Tween 80, is unknown. Tween-80, a very well-known non-ionic surfactant, belongs to the family of polyoxyethylene sorbitan fatty acid esters. Furthermore, it is commonly used in various industries, including pharmaceuticals, food, and cosmetics, due to its emulsifying, solubilizing, and dispersing properties [12,13,14].

Kappa-carrageenan (k-C) is a natural polysaccharide commonly found in certain species of red algae, such as *Eucheuma cottonii* and *Kappaphycus alvarezii*. This compound has been widely used in the food and cosmetic industries due to its gelling and thickening properties. k-C is known for its ability to form solid and resilient gels in the presence of metal ions. This characteristic makes it a valuable ingredient in manufacturing food products such as desserts, ice creams, dairy items, and processed foods, which enhance the texture and stability. In addition to its applications in the food industry, kappa-carrageenan is also employed in cosmetic and pharmaceutical products, such as gels and creams, due to its capacity to retain water and improve product consistency [15,16,17]. Its use combined with PC has increased the viscoelastic properties and viscosity of the aqueous phase studies compared to PC systems [18]. Nevertheless, this combination has never been studied in emulsion systems.

Avocado oil is primarily composed of monounsaturated fats, particularly oleic acid. These fats can help reduce the risk of heart disease by lowering LDL (low-density lipoprotein) cholesterol levels. Furthermore, it is a good source of various nutrients, including vitamin E, vitamin K, and folate. Vitamin E, in particular, is an antioxidant that can help protect cells from oxidative damage [19]. It also possesses anti-inflammatory properties due to the occurrence of phytosterols. These can be beneficial for reducing inflammation in the body and potentially relieving symptoms of inflammatory conditions. Hence, its use in food products can promote health benefits.

The main objective of this work was to obtain nanoemulgels mainly using compounds obtained from algae such as PC and k-carrageenan since the information on the combined use of these two is very limited. The mechanism of nanoemulgels involves the emulsification of the aqueous and oily phases, followed by the gelation of the matrix to form a stable structure with nano-scale particles dispersed in it. An optimization of the number of cycles using a microfluidizer and the ratio of PC and a well-known synthetic surfactant (Tween 80) was carried out via response surface methodology. Then, another algae compound (k-C) was used as a thickener to reduce the creaming that these systems suffered. This study proves that the combination of two sustainable compounds based on algae (PC and k-C) have emerged as promising candidates to stabilize dispersed systems for the emulsion industry.

## 2. Materials and Methods

### 2.1. Materials

The continuous phase was formulated using phycocyanin (PC), purchased from Naturegrail (Spain), Tween-80, provided by Merck^®^, and deionized water. The dispersed phase was formed by avocado oil. The dispersed phase consists of avocado oil and lemon essential oil, both supplied by Bidah Chaumel (Spain), in a 9:1 ratio. Kappa-carrageenan was kindly supplied by CP Kelco (Atlanta, GA, USA).

### 2.2. Emulgels Preparation

The aqueous phase was prepared by using different ratios of PC/Tween 80 (1/0; 3/1; 1/1; 1/3; and 0/1) in deionized water at a pH of 2.5. The oil phase consisted of avocado oil at 10 wt%. The selection of the optimum ratio of PC/Tween was conducted in the first part of this study. Batches of 250 g of these emulsions were prepared by primary homogenization using a Silverson L5M rotor-stator at 7000 rpm for 90 s. The secondary homogenization was also optimized in the first part of this study. Different systems were developed at 5000, 10,000, 15,000, 20,000, and 25,000 psi and at different numbers of cycles using a Microfluidizer^®^ M110P (Microfluidics, Shirley, NY, USA). This equipment was based on microchannel technology. Then, the nanoemulsion with optimized values of the PC/Tween ratio and the number of passes was used as the starting point for the addition of k-carrageenan to form nanoemulgels. A primary dispersion containing 0.6% by weight of the stabilizer was prepared by heating the required amount of water to boiling point, then adding the K-C, dispersing at 200 rpm for 5 min, and then cooling. This solution was then kept at 5 °C for 48 h to allow for the complete hydration of the polysaccharide. The final nanoemulgels were prepared by mixing the primary dispersion and the optimal nanoemulsion using the Ikavisc MR-D1 at 400 rpm for 30 min. Final nanoemulgels were obtained with different concentrations of kappa-carrageenan (0.1, 0.2, and 0.3 wt.%).

### 2.3. Statistical Analysis and Optimization Using Response Surface Methodology

In this study, we used design of experiments and response surface methodology (RSM) for the statistical analysis of our experimental data. RSM is a powerful and efficient tool for optimizing processes and understanding the relationship between multiple variables [20,21]. The central idea behind RSM is to model the response of a system as a function of multiple input variables, allowing researchers to explore the optimal conditions within the experimental domain. The data obtained were subjected to rigorous statistical analysis using Echip statistical software (Experimentation by design, Wilmington, DE, USA). The design of experiments in this study was carried out using Echip software with two numerical (type) factors: number of cycles (C) and the PC/Tween 80 ratio (R). The number of cycles was chosen as a factor in 10 levels from 1 to 11 cycles and the surfactant ratio (R) in 3 levels from 0 to 100% (100% means that the only emulsifier is Tween 80, while 0% means that the only emulsifier is phycocyanin), resulting in 35 designed experiments (including 2 replicates of the central point). The fitting of the quadratic model and subsequent analysis of variance (ANOVA) allowed us to assess the significance of each factor and their interactions, by following the methodology used by Khormali and Ahmadi (2023) [20]. The adequacy of the model was assessed using diagnostic plots and statistical metrics. The confidence level was 95%. The 3D response surface plots were generated to visualize the relationships between variables and to identify regions of interest for optimization. Numerical optimization techniques were applied using Echip to determine the optimum conditions that minimized the response variables: mean diameter and span.

### 2.4. Droplet Size Distributions

The droplet size distributions of the PC/Tween 80-based emulsions were measured by means of a Malvern Mastersizer 2000 (Malvern Panalytical, Malvern, UK), which uses laser diffraction technology. Three measurements were made for each sample. Finally, two different parameters were evaluated: the volumetric mean diameter (D_4,3_) and span. The former was used to quantify the mean droplet sizes:(1)$$D_{4,3}=\frac{\sum_{i=1}^{N}n_{i}d_{i}^{4}}{\sum_{i=1}^{N}n_{i}d_{i}^{3}}\;,$$
where d_i_ is the droplet diameter, n_i_ is the number of droplets of diameter d_i,_ and N is the total number of droplets. The latter was analyzed to quantify the polydispersity of the droplet size distributions:(2)$$Span=\frac{d\left(v,0.9\right)-d(v,0.1)}{d(v,0.5)},$$
where d(v,0.9), d(v,0.5), and d(v,0.10) are the diameters at 90%, 50%, and 10% cumulative volume.

### 2.5. Rheological Characterization

A rheological study was performed using an AR2000 controlled stress rheometer (TA Instruments, New Castle, DE, USA). The nanoemulgels formed with different concentrations of k-carrageenan were measured using a serrated plate–plate geometry (60 mm diameter, 1 mm gap). Small Amplitude Oscillatory Shear (SAOS) tests were carried out in terms of stress and frequency sweeps. The frequency sweeps were conducted in the linear viscoelastic range of the samples. In addition, flow tests were performed using a multi-step protocol (3 min/point). The equilibration time (the time prior the test) was 3 min and all the measurements were conducted at 20 °C.

### 2.6. Physical Stability

The destabilization processes that can suffer the emulsions and emulgels developed were analyzed by means of Turbiscan Lab Expert (Formulaction, Toulouse, France). The samples were introduced into measuring cells and kept at room temperature. Measurements of backscattering were carried out at different aging times as a function of the height of the measuring cell. The results are shown as backscattering (BS) with height as a function of the aging time. In addition, the Turbiscan Stability Index (TSI) was also calculated as follows:(3)$$TSI=\sum_{j}\left|{scan}_{ref}\left(h_{j}\right)-{scan}_{i}\left(h_{j}\right)\;\right|\;$$
where scan_ref_ and scan_i_ are the initial value of the backscattering and the value at a given time, respectively, and h_j_ is a given height of the measuring cell. This parameter allows different destabilization processes to be compared in terms of stability. Higher values of TSI mean poorer physical stability.

## 3. Results and Discussion

### 3.1. Preliminary Results for Avocado/Lemon Oil Emulsions

Figure 1 shows the droplet size distribution for emulsions formulated with PC and Tween 80 in a ratio of 1:1 as a function of the homogenization pressure. It is important to note that this ratio was chosen by way of an example. Firstly, a great reduction in the droplet size is observed from the pre-emulsion (emulsion processed with only primary homogenization) to 5000 psi. In this way, the use of microfluidization is a key parameter to reduce the droplet size for emulsions with PC/Tween. Then, there is a decrease in the droplet size with the homogenization pressure without a tendency to stabilize, reaching the lowest value of volumetric mean diameters at 25,000 psi (see Table 1).

### 3.2. Optimization of the Formulation and Processing for Avocado/Lemon Oil Emulsions

The droplet size distribution (DSD) for emulsions formulated with the PC/Tween 80 mixture as an emulsifier as a function on a number of passes through the microfluidizer is observed in Figure 2, by way of an example. As mentioned above, there is a huge reduction in the droplet size using the microfluidization technique. This fact is also shown in other studies about nanoemulsions [2,22]. A large reduction in the size and polydispersity is also observed when going from one to two passes. Interestingly, it seems that the peak is narrower with the number of passes. This fact indicates that the emulsifiers cover the interface efficiently and recoalescence does not occur due to a possible over-processing [23,24]. The trends observed for samples containing only each of the emulsifiers, PC or Tween 80, showed the same behavior as in the case of the mixture.

Irrespective of the emulsifier used, the mean droplet diameter and span decrease as the number of passes increases until it reaches a value at which they become approximately constant (see Table 2). The results illustrate that, within the systems studied, the best results are obtained for the surfactant Tween 80. Moreover, after 6–8 passes through the high-pressure system, the values of the diameters and span do not vary significantly. It should also be noted that for both the mixture of the algae extract and Tween 80 and the synthetic surfactant alone, monodisperse nanoemulsions can be developed with span values of less than one.

The results of the volumetric diameter and span values were analyzed using response surface methodology. Figure 3 and Figure 4 indicate the relationship between the (A) volumetric mean diameter (D_4,3_) and (B) span values with the number of cycles and the PC/Tween ratio. The volumetric mean diameter and the span values were fitted with a quadratic function of the number of cycles (C) and the ratio of PC/Tween (R):(4)$$D_{4,3}\left(\mu m\right)=0.119-0.018\cdot R-0.019\cdot C+0.008\cdot R^{2}+0.024\cdot C^{2}$$
(5)$$Span=0.761-0.249\cdot R-0.281\cdot C+0.087\cdot R\cdot C+0.209\cdot R^{2}+0.389\cdot C^{2}$$

The values of both R^2^ (coefficient of determination) were 0.90 and 0.89, which indicate a good correlation between the measured results and the predicted ones. On the other hand, the adjusted R^2^ values were 0.88 and 0.87, respectively. No significant lack of fit (F_crit_ > F_lof_, with *p* = 0.05) was observed, proving that the models used were adequate. In addition, the values of R^2^ and adjusted R^2^ are very high and comparable, indicating that the obtained quadratic model for predicting the mean diameter and polydispersity of the samples provides sufficient information to adequately describe the experimental data. In this sense, the models predicted that both D_4,3_ and the span values were influenced by both studied variables, the number of cycles, and the ratio of PC/Tween 80. There is a decrease in the volumetric diameter with the number of passes until a bit above the seventh cycle and with the Tween concentration. Hence, the seventh pass was proved to be the optimum to reduce the droplet size. Concerning the PC/Tween 80 ratio, the results obtained with the 1:1 ratio and with only Tween 80 as the emulsifier are not significantly different. On the other hand, the minimum value of the span was obtained using a 1:1 ratio and seven cycles through the Microfluidizer 110P. Because of the abovementioned and in order to minimize the use of synthetic emulsifiers, the ratio of 1:1 and the number of passes, seven, were chosen to be studied in the further section. Furthermore, in order to check the adequacy of the model, verification experiments by triplicate were carried out in the optimum conditions: seven cycles at 50/50 ratio. The mean D_4,3_ of these nanoemulsions was 0.124 ± 0.016 μm, which was close to the predicted value (0.106 ± 0.012 μm).

Furthermore, Figure 5 shows the backscattering (BS) with the height of the measuring cell as a function of the aging time for the selected system (seven cycles, 1:1 ratio). A decrease in the BS with the aging time in the low zone of the measuring cell was observed. This fact is related to the clarification of the low zone due to a creaming process. Because of that, the incorporation of a thickener (k-carrageenan), which could retard the destabilization phenomenon, was studied in the further section. No coalescence destabilization or Ostwald ripening mechanisms were observed, as evidenced by the zero BS variation in the intermediate zone of the sample and verified by the laser diffraction measurements.

### 3.3. Influence of k-Carrageenan on Rheological Properties and Physical Stability for Avocado/Lemon Oil Emulsions

Figure 6 illustrates the flow behavior of emulsions with no k-carrageenan and with 0.1, 0.2, and 0.3 wt.% of k-carrageenan. The emulsion formulated without thickener shows a Newtonian behavior (Newtonian viscosity = 3.88 mPa·s), typical of a non-structured system. On the other hand, shear-thinning flow behavior with an apparent zero-shear viscosity was observed for emulsions formulated with k-carrageenan, regardless of its concentration. This fact is related to the more structured systems. These data were fitted to the Cross model (R^2^ < 0.98), Equation (6).
(6)$$ƞ=\frac{{ƞ}_{0}-{ƞ}_{\infty }}{1+{\left(k\dot{\gamma }\right)}^{1-n}}$$
where ƞ is the viscosity, 𝛾̇ is the shear rate, ƞ_0_ is the viscosity at very low shear rates, ƞ_∞_ is the viscosity at very high shear rates, k is the inverse of the critical shear rate, and n is the flow index. The results of the fitting are shown in Table 3.

In addition, there is an increase in the zero-shear viscosity with the k-carrageenan concentration, as expected [25,26]. The values of the flow index (very close to zero) suggest high structuration grades, which can be related to long physical stabilities against creaming [3,27].

Figure 7 shows the mechanical spectra for emulsions using the optimized parameters (1:1 ratio PC/Tween, seven cycles) as a function of the k-carrageenan concentration. The values of the elastic modulus (G′) were higher than the viscous one (G″) for all systems in the range studied. This fact is a gel-type behavior, suggesting the creation of a 3D network for these systems. Previously, it has been said that k-carrageenan could enhance the network structure and weaken the frequency dependence in PC dispersions [18]. Hence, this behavior could be applied to emulsions systems based in PC. Furthermore, the gel structure confers to the system’s high stability against destabilization mechanisms such as creaming and coalescence. As expected, higher values of G′ and G″ were observed with higher concentrations of k-carrageenan.

Figure 8 shows the Turbiscan Stability Index (TSI) values for all nanoemulsions and nanoemulgels as a function of the k-carrageenan concentration. The nanoemulsion formulated without k-carrageenan showed a higher TSI (lower stability) than those prepared with the biopolymer. Therefore, the reason for the high TSI values of the nanoemulsion without thickener was due to the low viscosity described above. This supports the rheology results discussed above.

In fact, the most stable nanoemulsions were those prepared with 0.2 and 0.3 wt.%. This excellent result, in terms of physical stability, can be explained by the fact that the nanoemulgels prepared with these concentrations of biopolymers had sufficient viscosity to prevent creaming. The incorporation of only 0.2 or 0.3 wt.% of k-C is enough to provoke the presence of a gel-type behavior and to stabilize the systems developed. In comparison with other systems [28,29], 0.2 wt.% is a low concentration for a thickener. It should be noted that there are no significant differences between the average droplet diameters as well as the polydispersity of the nanoemulgels and those of the corresponding nanoemulsions. Finally, it should be noted that the droplet sizes obtained for these nanoemulgels are similar or even smaller than others described in the literature with similar formulations [10,30].

## 4. Conclusions

Firstly, the preliminary study for determining the suitable homogenization pressure revealed that the best results in terms of the droplet size distribution were those obtained for the emulsion processed at the highest homogenization pressure (25,000 psi). Secondly, the study of the optimization of the PC/Tween 80 ratio and number of cycles through the microfluidizer via response surface methodology pointed out that the 1:1 ratio and the seventh pass as the minimum point of the volumetric diameter and span. However, creaming was observed in this system, using the MLS technique. As a consequence, in the third part of this study, k-carrageenan was incorporated as a thickener to retard this destabilization process. Interestingly, the presence of this biopolymer provoked pseudoplastic behavior with a very low value of the flow index. This fact is directly related to the occurrence of a strong network that can reduce the migration of the droplets. It is also proved by the presence of a gel-type system, observed by the SAOS tests. On top of that, the values of the TSI corroborated that the creaming process has been reduced thanks to the combination of k-carrageenan and PC. Hence, stable nanoemulgels (D_4,3_ = 124 nm) formulated with compounds obtained from algae (PC and k-C) were developed. This work brings to light that the combination of PC and k-C is a great way to stabilize dispersed systems and to obtain nanoemulgels with potential applications in the food industry. As is our understanding, this work contributes to the knowledge about PC and its use in combination with other polymers, such as k-carrageenan.

## Figures and Tables

**Figure 1 polymers-16-00194-f001:**
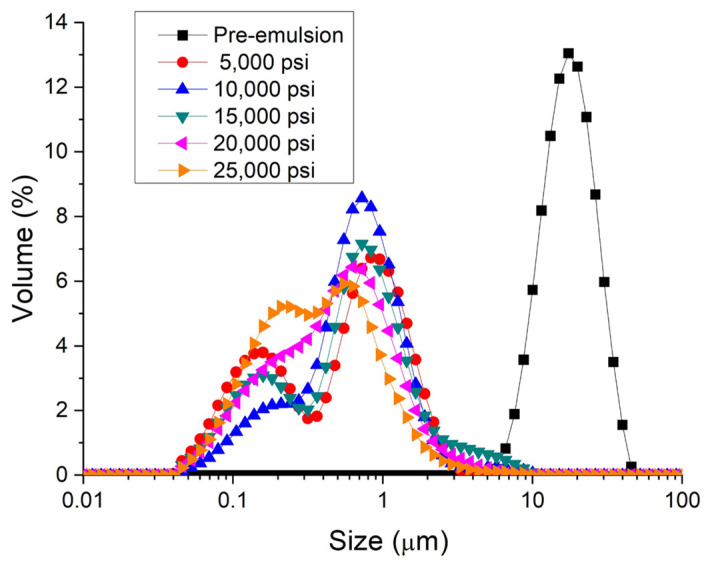
Droplet size distributions for emulsions containing phycocyanin and Tween 80 as emulsifiers (ratio 1:1) as a function of processing.

**Figure 2 polymers-16-00194-f002:**
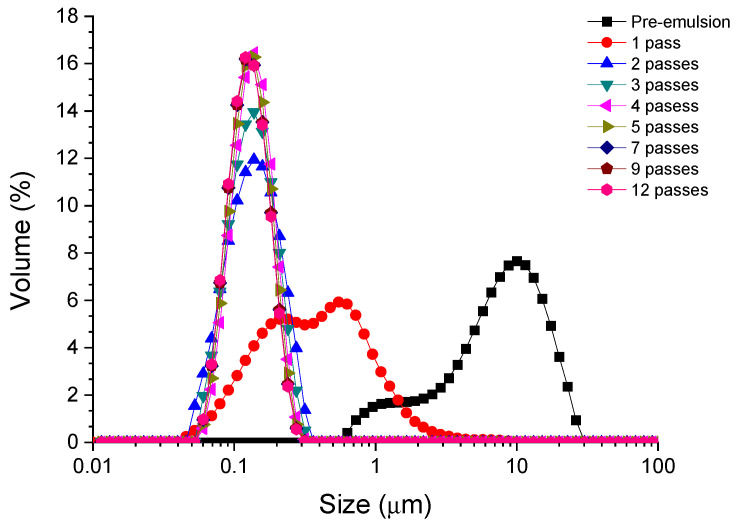
Droplet size distributions for emulsions containing a mixture of phycocyanin and Tween 80 (ratio 1:1) as emulsifiers.

**Figure 3 polymers-16-00194-f003:**
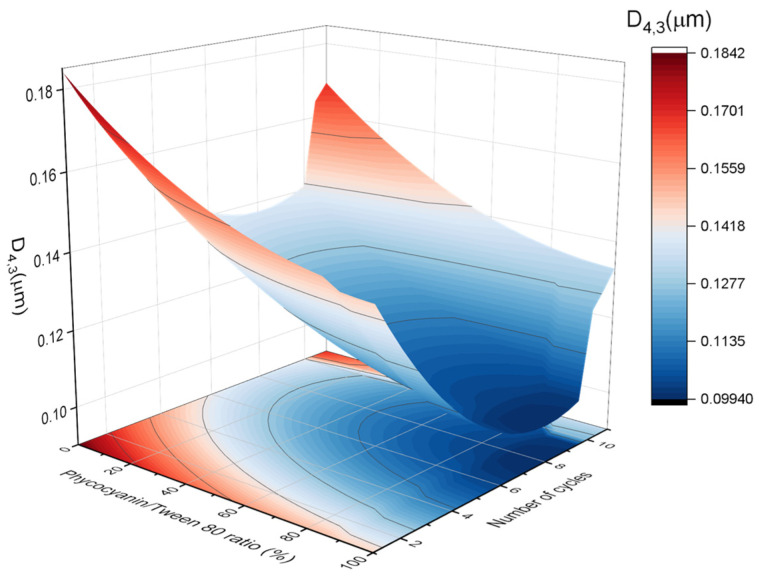
Response surface 3D plot of volumetric mean diameter (D_4,3_) as a function of the number of cycles and the ratio of PC/Tween 80. A figure of 100% means that the only emulsifier is Tween 80 while 0% means that the only emulsifier is phycocyanin.

**Figure 4 polymers-16-00194-f004:**
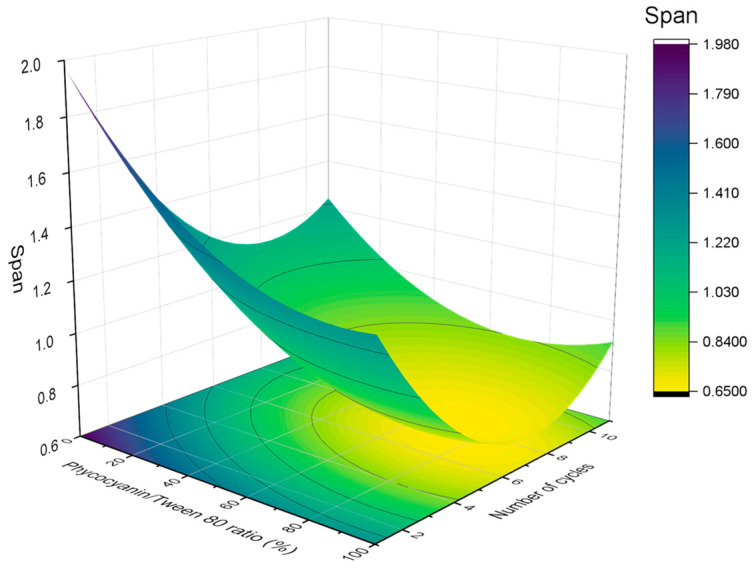
Response surface 3D plot of span values as a function of the number of cycles and the ratio of PC/Tween 80. A figure of 100% means that the only emulsifier is Tween 80 while 0% means that the only emulsifier is phycocyanin.

**Figure 5 polymers-16-00194-f005:**
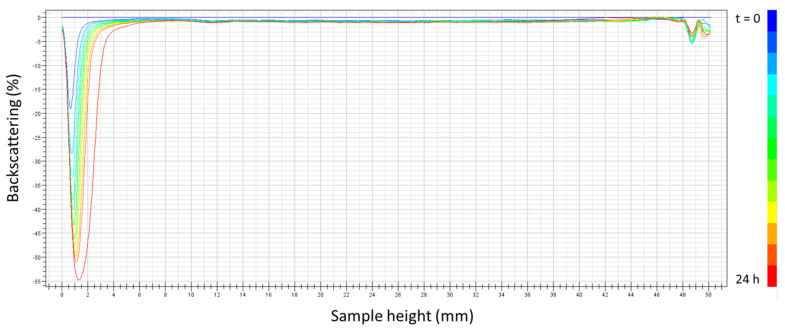
Backscattering with measuring cell height as a function of aging time for the emulsion prepared using optimized conditions (7 cycles, 1:1 ratio PC/Tween).

**Figure 6 polymers-16-00194-f006:**
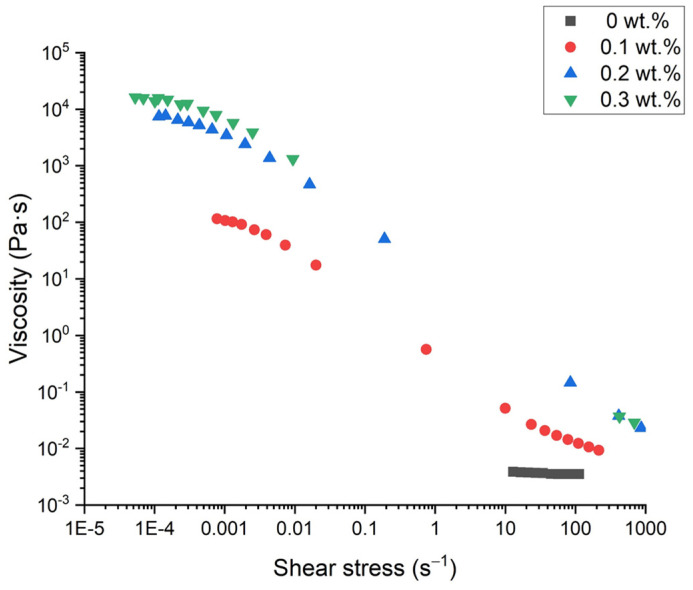
Flow curves for emulsions with optimized parameters as a function of k-carrageenan concentration at one day of aging time.

**Figure 7 polymers-16-00194-f007:**
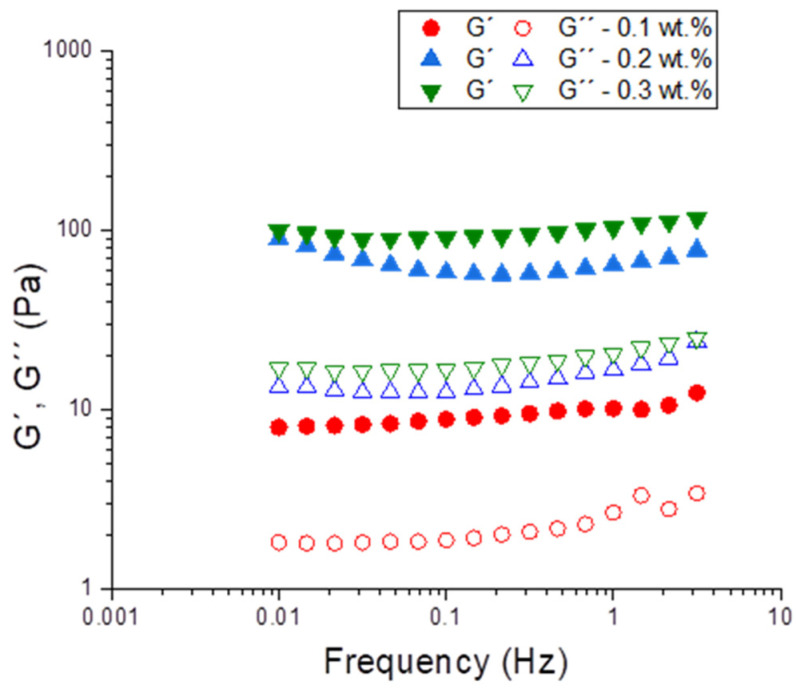
Frequency sweeps for emulsions using optimized parameters as a function of k-carrageenan concentration.

**Figure 8 polymers-16-00194-f008:**
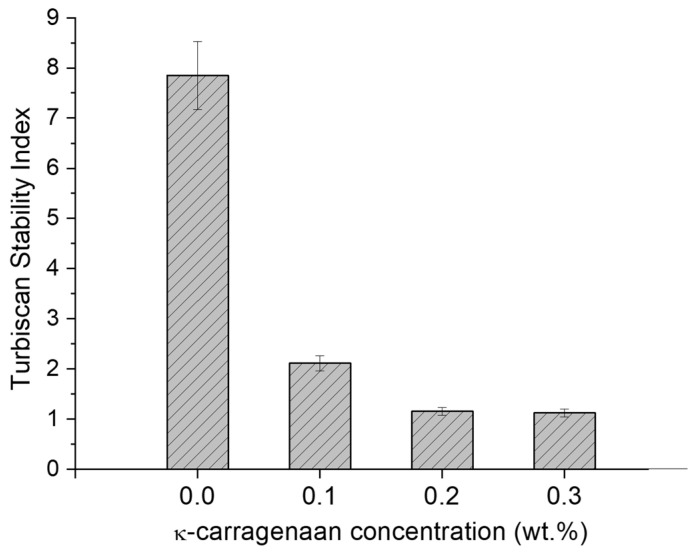
Turbiscan Stability Index values at 14 days for the nanoemulsions and nanoemulgels as a function of k-carrageenan concentration. Temperature = 20 °C.

**Table 1 polymers-16-00194-t001:** Volumetric mean diameter (D_4,3_) and span for emulsions containing phycocyanin and Tween 80 as emulsifiers (ratio 1:1) as a function of processing.

Sample	D_4,3_ (μm)	Span
Pre-emulsion	17.413	1.073
5000 psi	0.855	2.633
10,000 psi	0.771	1.917
15,000 psi	0.663	2.310
20,000 psi	0.629	2.511
25,000 psi	0.510	2.668

**Table 2 polymers-16-00194-t002:** Volumetric mean diameter (D_4,3_) and span for emulsions containing phycocyanin, a mixture of phycocyanin/Tween 80, or Tween 80 as emulsifiers as a function of microfluidization passes.

Number of Passes	Phycocyanin		Mixture		Tween 80	
	D_4,3_ (μm)	Span	D_4,3_ (μm)	Span	D_4,3_ (μm)	Span
0	13.810	1.576	8.132	1.981	7.432	1.559
1	0.688	2.972	0.510	2.668	0.422	1.985
2	0.176	1.706	0.136	1.139	0.131	0.967
3	0.162	1.508	0.132	0.982	0.123	0.829
4	0.162	1.537	0.131	0.836	0.119	0.824
5	0.148	1.361	0.128	0.847	0.118	0.821
6	0.147	1.325	0.124	0.845	0.115	0.817
7	0.147	1.204	0.124	0.844	0.112	0.814
8	0.145	1.195	0.124	0.844	0.112	0.812
9	0.145	1.177	0.124	0.845	0.111	0.812
10	0.145	1.177	0.124	0.842	0.109	0.807
11	0.145	1.176	0.124	0.844	0.109	0.806
12	0.145	1.176	0.123	0.842	0.110	0.818

**Table 3 polymers-16-00194-t003:** Cross model fitting parameters. The values were obtained by using a non-linear regression analysis.

k-Carrageenan Concentration (wt.%)	η_0_ (Pa·s)	η_∞_ (Pa·s)	k (s)
0.1	115	<0.001	237
0.2	7434	<0.001	1154
0.3	16,170	<0.001	1394

## Data Availability

The data presented in this study are available on request from the corresponding author.

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
