# Peer review of "Formulation and Characterization of Sustainable Algal-Derived Nanoemulgels: A Green Approach to Minimize the Dependency on Synthetic Surfactants"

_polymers, 2024, doi:10.3390/polym16020194_

Round 1
Reviewer 1 Report
Comments and Suggestions for Authors
The experimental article “Formulation and characterization of sustainable algal-derived nanoemulgels: a green approach to minimize the dependency on synthetic surfactants” is devoted to the conditions for obtaining nanoemulgels based on the natural proteins phycocyanin (PC) and k-carrageenan. Both substances are isolated from algae. PC is characterized by powerful antioxidant and anti-inflammatory properties, therefore it is used in medicine and the food industry. The k-carrageenan polymer is known as a thickener. According to the authors, the combined use of these two environmentally friendly and algae-derived polymers provided stable nanoemulgels for the food industry. The article has a very small volume, since this area is actively published by the authors (references 9, 10 and 17), the list of references is only 17 articles, their use is justified. The authors used the surface response methodology, described in detail the progress of their research and illustrated their results. The article may be published after some comments and questions have been resolved.
Comments and questions:
1. In the introduction, the authors set out the purpose of the research in great detail. But the information provided in the Conclusion does not reflect the achievement of this goal and the question of the scientific novelty of the results obtained remains open. This situation needs to be corrected.
2. By eliminating remark 1, the authors will probably be able to improve the abstract, focusing readers’ attention precisely on the scientific novelty of their results, and not on the origin of the two polymers.
3. What is the droplet diameter for nanoemulgels and how do the results compare with published data?
4. Lines 218-229. The authors found that 7 cycles of using Microfluidizer 110P are necessary. Are these data correct only for a specific emulgel and cannot be predicted for emulgels of other compositions?
5. Carrageenan is a well-known thickener, so why do the authors consider the results obtained when using carrageenan at a concentration of 0.2 and 0.3 wt% to be “excellent”?
Author Response
The experimental article “Formulation and characterization of sustainable algal-derived nanoemulgels: a green approach to minimize the dependency on synthetic surfactants” is devoted to the conditions for obtaining nanoemulgels based on the natural proteins phycocyanin (PC) and k-carrageenan. Both substances are isolated from algae. PC is characterized by powerful antioxidant and anti-inflammatory properties, therefore it is used in medicine and the food industry. The k-carrageenan polymer is known as a thickener. According to the authors, the combined use of these two environmentally friendly and algae-derived polymers provided stable nanoemulgels for the food industry. The article has a very small volume, since this area is actively published by the authors (references 9, 10 and 17), the list of references is only 17 articles, their use is justified. The authors used the surface response methodology, described in detail the progress of their research and illustrated their results. The article may be published after some comments and questions have been resolved.
Many thanks for your extremely useful comments. The references have been extended to 30.
Comments and questions:
- In the introduction, the authors set out the purpose of the research in great detail. But the information provided in the Conclusion does not reflect the achievement of this goal and the question of the scientific novelty of the results obtained remains open. This situation needs to be corrected.
In order to correct this point, a new paragraph has been introduced in the conclusions:
“Hence, stable nanoemulgels (D4,3= 124 nm) formulated with compounds obtained from algae (PC and k-C) were developed. This work brings to light that the combination of PC and k-C is a great way to stabilize dispersed systems and to obtain nanoemulgels with potential applications in food industry.”
- By eliminating remark 1, the authors will probably be able to improve the abstract, focusing readers’ attention precisely on the scientific novelty of their results, and not on the origin of the two polymers.
Many thanks for your useful comments. However, the journal encourages authors to use the following style of structured abstracts, but without headings: (1) Background: Place the question addressed in a broad context and highlight the purpose of the study; (2) Methods: briefly describe the main methods or treatments applied; (3) Results: summarize the article’s main findings; (4) Conclusions: indicate the main conclusions or interpretations. Hence, we cannot eliminate the origin of the two polymers that is part of the background. Nevertheless, we have incorporated the following lines to the abstract in order to focus readers’ attention on the scientific novelty of the results:
“This work has proved that the combined used of PC and k-C have emerged as a sustainable alternative to stabilize dispersed systems for the food industry.”
- What is the droplet diameter for nanoemulgels and how do the results compare with published data?
There are no significant differences in the mean droplet diameter of these nanoemulgels compared to nanoemulsions prepared and formulated under the same conditions (with the exception of the biopolymer). In any case, the following sentence has been added to the revised version of the text:
“It should be noted that there are no significant differences between the average droplet diameters as well as the polydispersity of the nanoemulgels and those of the corresponding nanoemulsions. Finally, it should be noted that the droplet sizes obtained for these nanoemulgels are similar or even smaller than others described in the literature with similar formulations.”
- Lines 218-229. The authors found that 7 cycles of using Microfluidizer 110P are necessary. Are these data correct only for a specific emulgel and cannot be predicted for emulgels of other compositions?
These data are specific to this formulation. If we change the formulation, the results of the optimization of processing parameters can vary.
- Carrageenan is a well-known thickener, so why do the authors consider the results obtained when using carrageenan at a concentration of 0.2 and 0.3 wt% to be “excellent”?
They are excellent results since from the stability point of view, the incorporation of only 0.2 or 0.3 wt% of k-C is enough to provoke the presence of a gel-type behaviour and to stabilise the systems developed. In comparation with other systems, 0.2 wt% is a low concentration for a thickener.
In order to clarify this, we have incorporated the following sentences in line 313:
“The incorporation of only 0.2 or 0.3 wt.% of k-C is enough to provoke the presence of a gel-type behaviour and to stabilise the systems developed. In comparation with other systems [28,29], 0.2 wt% is a low concentration for a thickener.”
Refs:
[28] Ren, Z., Li, X., Ma, F., Zhang, Y., Hu, W., Khan, M. Z. H., & Liu, X. (2022). Oil-in-water emulsions prepared using high-pressure homogenisation with Dioscorea opposita mucilage and food-grade polysaccharides: Guar gum, xanthan gum, and pectin. LWT, 162, 113468.
[29] Kumar, Y., Roy, S., Devra, A., Dhiman, A., & Prabhakar, P. K. (2021). Ultrasonication of mayonnaise formulated with xanthan and guar gums: Rheological modeling, effects on optical properties and emulsion stability. LWT, 149, 111632.

Reviewer 2 Report
Comments and Suggestions for Authors
The review of the manuscript entitled: “Formulation and characterization of sustainable algal-derived nanoemulgels: a green approach to minimize the dependency on synthetic surfactants”. The work is well-written in which the nanoemulgels were formulated with PC and k-carrageenan. By responding to the following comments and questions, the work can be ready for publication:
1. What is the mechanism of the used nanoemulgels? Please add it in the abstract.
2. Please add the main findings in the abstract. Also, RSM results and optimization should be presented.
3. The main properties and compositions of the Tween-80.
4. It is recommended to mention the model development and optimization by response surface method. The next references can be used:
https://doi.org/10.1007/s13202-023-01679-2
5. The variety of the concentration for the emulgels is not high (0.1, 0.2 and 0.3 wt.%).
6. R2, R2 adjusted and error analysis of RSM should be analyzed.
7. Can it be possible to observe the synergistic effect for the mixture of Table 2?
8. Please support the findings and results with references.
9. The number of references should be extended.
Author Response
· Reviewer 2
The review of the manuscript entitled: “Formulation and characterization of sustainable algal-derived nanoemulgels: a green approach to minimize the dependency on synthetic surfactants”. The work is well-written in which the nanoemulgels were formulated with PC and k-carrageenan. By responding to the following comments and questions, the work can be ready for publication:
- What is the mechanism of the used nanoemulgels? Please add it in the abstract.
Thanks for the suggestion. The mechanism of nanoemulgels has been included in the introduction section due to space limitations in the abstract:
The mechanism of nanoemulgels involves emulsification of the aqueous and oily phases, followed by gelation of the matrix to form a stable structure with nano-scale particles dispersed in it.
- Please add the main findings in the abstract. Also, RSM results and optimization should be presented.
We totally agree with your suggestions. The abstract has been modified as follows:
“Phycocyanin (PC), a natural protein which is very interesting from the medical point of view due to its potent antioxidant and anti-inflammatory properties, is obtained from algae. This compound is gaining positions for applications in food industry. The main objective of this work was to obtain nanoemulgels formulated with PC and k-carrageenan (a polymer that is obtained from algae as well). An optimization of the processing parameters (homogenization pressure and number of cycles) and the ratio of PC and a well-known synthetic surfactant (Tween 80) was developed using surface response methodology. The results of this optimization were 25000 psi, 7 cycles and 1:1 ratio PC/Tween. However, the necessity of the incorporation of a polymer which plays a thickener role was observed. Hence, k-carrageenan (k-C) was used to retard the creaming process that these nanoemulsions suffered. The incorporation of this biopolymer provoked the creation of a network that show gel-type behavior and flow indexes very closed to zero. Thanks to the combined used of these two sustainable and algae-obtained compounds, stable nanoemulgels were obtained. This work has proved that the combined use of PC and k-C have emerged as a sustainable alternative to stabilize dispersed systems for the food industry.”
- The main properties and compositions of the Tween-80.
The following paragraph has been incoroporated to the introduction part:
“Its use with synthetic surfactant, like Tween 80 is unknown. Tween-80, a very well-known non-ionic surfactant, belongs to the family of polyoxyethylene sorbitan fatty acid esters. Furthermore, it is commonly used in various industries, including pharmaceuticals, food, and cosmetics, due to its emulsifying, solubilizing, and dispersing properties.”
- It is recommended to mention the model development and optimization by response surface method. The next references can be used:
More information on the design of experiments and the RSM has been included, as well as two new references, including the one suggested by the reviewer.
The design of experiments in this study was carried out using Echip software with two numerical (type) factors: number of cycles (C) and the PC/Tween 80 ratio (R). The number of cycles was chosen as a factor in 10 levels from 1 to 11 cycles and the surfactant ratio (R) in 3 levels from 0 to 100% (100% means that the only emulsifier is Tween 80, while 0% means that the only emulsifier is phycocyanin), resulting in 35 designed experiments (including 2 replicates of the central point). The confidence level was 95%.
- The variety of the concentration for the emulgels is not high (0.1, 0.2 and 0.3 wt.%).
In order to assure a green approach to develop nanoemulgels with food applications, the concentration of the thickener (k-C) is not hight in order to reduce the compounds to the minimum. In addition, it is proved that these conconcetrations are enough to stabilised these nanoemulgels formulated with algae compounds.
- R2, R2-adjusted and error analysis of RSM should be analyzed.
A more in-depth statistical analysis of the results obtained has been included.
- Can it be possible to observe the synergistic effect for the mixture of Table 2?
We think that there is no synergistic effect, since the results of droplet sizes are not better than the results using only Tween-80 as surfactant. However, the droplet sizes for the seventh pass using the mixture in a ratio of 1:1 and using only Tween-80 as surfactant are not significantly different. Because of that, and in order to minimise the use of synthetic emulsifiers, the ratio of 1:1 (PC/Tween-80) was chosen.
- Please support the findings and results with references.
Many thanks for your suggestion. It is done.
- The number of references should be extended.
The number of references has been extended to 30.

Round 2
Reviewer 2 Report
Comments and Suggestions for Authors
Thanks for the revision of the manuscript. The work has improved. However, two comments were not completely responded as follows:
Comment 4: The model optimization by RSM and ANOVA can be described in detail. Also, the next suggested reference was not used for this purpose. Please use it: https://doi.org/10.1007/s13202-023-01679-2 (Khormali, A., & Ahmadi, S. (2023). Prediction of barium sulfate precipitation in dynamic tube blocking tests and its inhibition for waterflooding application using response surface methodology. Journal of Petroleum Exploration and Production Technology, 13(11), 2267-2281.)
Comment 5: The information for the short range of the concentration is not enough. please more explain
Please completely respond to the above comments.
In addition, on page 10, above figure 9, there is a non-English sentence. please remove or translate it
Author Response
Thanks for the revision of the manuscript. The work has improved. However, two comments were not completely responded as follows:
Thank you very much for your further revision which will help us authors to further improve the manuscript.
Comment 4: The model optimization by RSM and ANOVA can be described in detail. Also, the next suggested reference was not used for this purpose. Please use it: https://doi.org/10.1007/s13202-023-01679-2 (Khormali, A., & Ahmadi, S. (2023). Prediction of barium sulfate precipitation in dynamic tube blocking tests and its inhibition for waterflooding application using response surface methodology. Journal of Petroleum Exploration and Production Technology, 13(11), 2267-2281.)
Unfortunately there was confusion among the authors and the latest revised version of the manuscript was not attached. Thus, there was both a sentence in our language (Spanish) and missing references. Among them, the one cited by the reviewer is now cited both in materials and methods and at the end of the text.
Comment 5: The information for the short range of the concentration is not enough. please more explain.
Taking into account the principles of bioeconomy, one of the objectives of the work was the development of nanoemulgels with the minimum concentrations of bio-based components to be stable and with desirable rheological properties, in this case gel-like or weak gel behavior. From 0.2 wt.% stable systems are obtained, and with 0.3 wt.% the gel-weak rheological properties are obtained. Although it might be interesting to increase the concentration of kappa carrageenan (or even reduce it), we consider that above the range studied we would obtain systems very close to the solid state (jam-like, compared to other food systems) and below it we would not obtain sufficient physical stability. Hence our decision to only study this range of concentrations.
Please completely respond to the above comments.
In addition, on page 10, above figure 9, there is a non-English sentence. please remove or translate it
